# Associations between Maternal Dietary Patterns and Infant Birth Weight in the NISAMI Cohort: A Structural Equation Modeling Analysis

**DOI:** 10.3390/nu13114054

**Published:** 2021-11-12

**Authors:** Jerusa da Mota Santana, Valterlinda Alves de Oliveira Queiroz, Marcos Pereira, Enny S. Paixão, Sheila Monteiro Brito, Djanilson Barbosa dos Santos, Ana Marlucia Oliveira

**Affiliations:** 1Center of Health Sciences, Universidade Federal do Recôncavo da Bahia, Santo Antônio de Jesus, Avenida Carlos Amaral, R. do Cajueiro, 1015, Bahia 44574-490, Brazil; sheilambrito@ufrb.edu.br (S.M.B.); djanilson@ufrb.edu.br (D.B.d.S.); 2School of Nutrition, Universidade Federal da Bahia, Salvador 40110-150, Brazil; valterlinda.oliveira@gmail.com (V.A.d.O.Q.); anamarluciaoliveira@gmail.com (A.M.O.); 3Collective Health Institute, Universidade Federal da Bahia, Salvador 40110-040, Brazil; pereira.santosm@yahoo.com; 4Epidemiology and Population Health, London School of Hygiene and Tropical Medicine, London WC1E 7HT, UK; enny.cruz@lshtm.ac.uk

**Keywords:** pregnancy, infant, dietary patterns, birth weight, factor analysis, cohort, Brazil

## Abstract

The mother’s diet during pregnancy is associated with maternal and child health. However, there are few studies with moderation analysis on maternal dietary patterns and infant birth weight. We aim to analyse the association between dietary patterns during pregnancy and birth weight. A prospective cohort study was performed with pregnant women registered with the prenatal service (Bahia, Brazil). A food frequency questionnaire was used to evaluate dietary intake. Birth weight was measured by a prenatal service team. Statistical analyses were performed using factor analysis with a principal component extraction technique and structural equation modelling. The mean age of the pregnant women was 27 years old (SD: 5.5) and the mean birth weight was 3341.18 g. It was observed that alcohol consumption (*p* = 0.05) and weight-gain during pregnancy (*p* = 0.05) were associated with birth weight. Four patterns of dietary consumption were identified for each trimester of the pregnancy evaluated. Adherence to the “Meat, Eggs, Fried Snacks and Processed foods” dietary pattern (pattern 1) and the “Sugars and Sweets” dietary pattern (pattern 4) in the third trimester directly reduced birth weight, by 98.42 g (Confidence interval (CI) 95%: 24.26, 172.59) and 92.03 g (CI 95%: 39.88, 165.30), respectively. It was also observed that insufficient dietary consumption in the third trimester increases maternal complications during pregnancy, indirectly reducing birth weight by 145 g (CI 95%: −21.39, −211.45). Inadequate dietary intake in the third trimester appears to have negative results on birth weight, directly and indirectly, but more studies are needed to clarify these causal paths, especially investigations of the influence of the maternal dietary pattern on the infant gut microbiota and the impacts on perinatal outcomes.

## 1. Introduction

Birth weight is a significant factor in children’s health: the evidence indicates that both high or low birth weight are risk factors for the occurrence of infectious and respiratory diseases and can adversely affect growth and development in infancy. The inadequate nutritional status of children in the intrauterine environment also increases the possibility of the emergence of chronic diseases in later life, in particular obesity, excess weight, diabetes mellitus, metabolic syndrome, and other chronic diseases [1,2], thus indicating the association between low birth weight and specific micronutrient deficiencies during pregnancy [3].

The current dietary pattern of pregnant women is characterized by insufficient intake of fruit and vegetables, sources of fibre, and micronutrients, and a greater intake of processed and ultra-processed sources of saturated fats, trans fats, and added sugar [4,5,6,7,8,9,10,11]. This dietary pattern is a risk factor for inadequate gestational weight gain, with subsequent negative effects on birth weight [8,9]. Inadequate dietary intake during pregnancy increases the risk of maternal morbidity and mortality and affects the health and nutrition of the foetus and child at birth, increasing the prevalence of excess weight and obesity in infancy and adulthood [1,12,13].

Epidemiological studies reveal that women who adhered to a dietary pattern considered to be healthy during pregnancy (fruits, vegetables, grains, and fish) presented lower chances of having low-weight [14] and small newborns for the gestational age [15]. A recent systematic review with evidence of observational studies involving 167,507 women recorded that adhesion to unhealthy diets composed of refined grains, processed meat, and sources of saturated fat and sugar during pregnancy was associated with a lower birth weight and higher chance of prematurity [16]. This is because the characteristics of the prenatal maternal diet, such as daily fruits and the intake of red meat and dairy products, are linked to the composition of the infant microbiota, which may be associated with neonatal outcomes [17].

Although these studies have examined the determinants of birth weight, factors associated with moderation analysis on maternal dietary patterns in the first and third gestational trimesters and their relationship with birth weight remain unelucidated among socioeconomically vulnerable populations living in low- and middle-income countries, such as Brazil. It is thus postulated that adequate food consumption can reduce unfavourable results and risks during pregnancy and childbirth for both the mother and foetus, thereby offering protection to the intestinal health and nutritional state of the children, particularly with respect to birth weight. Thus, this study aims to identify the association between dietary patterns during pregnancy and birth weight.

## 2. Materials and Methods

### 2.1. Study Design and Sample Calculation

This is a longitudinal NISAMI (Núcleo de Investigação em Saúde Materno-Infantil, Centre for Research on Maternal–Infant Health, in English) study involving pregnant women residing in the urban area of the municipality of Santo Antônio de Jesus, in Bahia, Brazil, between April of 2012 and June of 2013.

Sample size calculation was based on an average weight at birth of 3196 g and a standard deviation of 456 g [18], assuming a difference of 100 g in birth weight when coupled with adequate dietary intake, a sample error of 2%, a power of 80%, and an acceptable loss of 15%. Thus, a sample size of 175 pregnant women would be needed. However, we decided to include in the sample all 185 pregnant women who were contacted in the cohort.

### 2.2. Exclusion and Inclusion Criteria 

The sample included pregnant women aged 19 or above who were resident and domiciled in the urban area of the municipality and registered with the prenatal service of the Family Health Units (FHUs), and who were in their first gestational trimester. 

The following women were considered ineligible for the study: pregnant women monitored at the prenatal service but who resided outside the municipality studied; adolescents; those with multiple pregnancies, stillbirths, and anembryonic pregnancies; those lacking monitoring in any of the rounds or pregnant women who were not found at their homes (Figure 1).

### 2.3. Data Collection and Dietary Intake Assessments

The pregnant women were recruited at the prenatal services of the FHUs of the municipality of Santo Antônio de Jesus, according to the inclusion criteria for entry into the research. The study was conducted in four rounds, the first three of which were carried out during pregnancy and the last of which occurred in the immediate postpartum period. The stages of monitoring the cohort were considered as rounds. The baseline (first round) occurred in the first trimester at the FHU and involved semi-structured interviews to meet the pregnant women and understand their socioeconomic conditions, demographics, reproductive and/or obstetric history, and lifestyle, with their answers being recorded in a questionnaire. Information about biochemical parameters, such as glycemia, hemogram, VDRL, HTLV, cytomegalovirus, rubella, and faeces parasitology, was obtained from the prenatal service patient records; the anthropometric assessment (height and weight) was carried out at the time of the interviews by suitably trained researchers and the dietary intake evaluation was conducted using the food frequency questionnaire (FFQ) [19]. The second and third rounds occurred, respectively, in the second and third trimesters of pregnancy at the woman’s home, when the anthropometric and dietary intake evaluations were carried out, the latter of which was only carried out in the third trimester. We decided not to collect dietary intake data in the second trimester of pregnancy to prevent loss to follow-up.

In the remaining rounds, this information was collected using a validated semi-quantitative food frequency questionnaire (FFQ). It consists of 74 food items that represent the local dietary culture and includes over 11 food groups (Appendix A).

The frequency of consumption was represented by the following categories: daily, weekly, monthly, never, or almost never [20]. The value of one was multiplied by the interval of the daily frequency reported. For the options of weekly and monthly intervals, we used the average of the range of frequencies divided by the period, whether weekly (=7) or monthly (=30), thus obtaining the average daily consumption. In all, 11 food groups were constructed (Appendix A). In order to estimate the size of the food portions consumed, a photographic record album of food rations and utensils was used. Data on food consumption was elaborated by the researcher [21].

### 2.4. Data on Study Variables

The details of the cohort procedures are described in previous studies [19]. In the first stage of the cohort (baseline), the information on socio-economic and demographic conditions, reproductive and/or obstetric history, and lifestyle were collected during the enrolment of pregnant women at the prenatal services, and responses were recorded using a questionnaire. 

We used the pre-pregnancy body mass index (PP-BMI) and classified it according to the Institute of Medicine parameters [20]. Maternal weight and height measurements were taken during the interview by trained researchers.

We used gestational BMI as a proxy for gestational anthropometric status, and this was classified according to the Atalah curve [22]. Weight gain during pregnancy was assessed according to the categories of PP BMI, following Institute of Medicine (IOM) recommendations [23]. A Marte^®^ portable digital scale with a capacity of 150 kg was used to record weight. Height was measured using a Welmy^®^ brand stadiometer with a 2000 mm maximum height. Anthropometric measurements were taken in duplicate. Maximum variations of 0.5 cm, for measuring height, and 100 g, for measuring weight, were accepted [24].

### 2.5. Outcomes

In the follow-up of the study (April 2012 and June 2013), the pregnant women were monitored until the birth of their child [19,20]. The birth weight was measured with the child naked using a digital paediatric scale (Welmy^®^), with a capacity of 15 kg and a 10 g interval. The anthropometric measurement was performed in duplicate, and a maximum variation of 10 g was acceptable. When different values were obtained, a third measurement was performed. The final measurement was calculated as the mean of the values of the closest measurements [24].

Information on birth weight was collected from the municipality’s Epidemiological Survey (VIEP) registration system. When necessary, home visits were performed at the end of the study for those pregnant women whose data on gestational outcomes were not found in the VIEP [25,26]. 

### 2.6. Data Management and Statistical Analyses

Prevalence was determined to describe the categorical variables. Mean and standard deviation (SD) were used for the continuous variables. The mean of birth weight according to the exposure variables was compared using Student’s *t* test for equal variances.

To identify the dietary patterns in the sample, factor analysis by principal component was used. Varimax rotation of the factors was employed, with graphic Scree plot used to define the extracted (retained) factors. A factorial charge ≥0.4 was adopted as the selection criterion for the dietary groups to be included in the pattern. Internal consistency of each factor was evaluated by Cronbach’s alpha. Detailed information on the identification of dietary patterns during pregnancy is reported elsewhere [27].

To evaluate the association between dietary patterns during pregnancy and birth weight, structural equation modelling (SEM) was used. In this sense, birth weight represented the response variable (endogenous variable); pattern of dietary consumption (constructed) represented the Exposure variables (exogenous variable); and complications during pregnancy, PP BMI, and weight gain (observable variables), intermediate variables. Associations were accepted with a *p* < 0.05. The variables (BMI, gestational BMI, weight gain) were included continuously into the SEM. Patterns of dietary consumption (constructed) were entered into the model as a factorial score in a continuous way.

We used the likelihood-ratio Chi-square statistical test to analyse the appropriateness of the model. The results of the analysis for the group of mediating variables were evaluated by means of standardized coefficients (SC) and interpreted according to the results of Kline [28].

Excel software programs were used for processing the dietary consumption data. The Statistical Package for Social Sciences program, version 17, was used for data entry and for descriptive and factor analyses. Stata software, version 12.0, was used for SEM analysis.

## 3. Results

### 3.1. Cohort Characteristics

The characteristics of the mothers and their newborns are presented in Table 1; there was a predominance of maternal age lower than 30 (71.89%) and a low level of education (85.9%). The prevalence of anthropometric status characterized by excess weight (overweight/obesity) was 44.0% in the pre-pregnancy period and 48.1% during pregnancy.

Participant loss over the 12 months of follow-up was 7.57% (*n* = 14). The most frequent reasons were those related to not finding the address provided by the pregnant woman to the family clinic (6; 25%), absence in one of the rounds of the study (8, 33.4%), and refusal to participate in the study (10; 41.6%) (Figure 1).

The prevalence of adequate weight at birth was 96.2% (average, 3341.18 g; SD: 522.832) (data not presented in the tables). The highest mean birth weight was found for infants born to mothers with family income of more than one minimum salary (3371.67 SD: 508.31) and for those whose mothers had more than seven medical appointments in the prenatal period (3376.87 g; SD: 440.364), had no complications during pregnancy (337 6.87; SD:528.30), and with more than 10 kg weight gain during pregnancy (3424.36; SD: 490.30).

### 3.2. Dietary Patterns during Pregnancy

In the first and third trimesters, four dietary consumption patterns were identified (Figure 2). The internal consistency of each pattern was assessed using Cronbach’s Alpha. All extracted patterns presented an acceptable Cronbach index (>0.5), indicating homogeneity among the constructs (Table 2).

The first pattern of the first trimester (processed foods/sugar/sweets, coffee, and fats) represented 19.42% of the consumption data; the second (legumes, vegetables, meat, and eggs) accounted for 11.23%; the third (cereals/roots/tubers, milk/dairy, and fried snacks), 10.68%; and the fourth (fruit), 9.11% of the consumption data (Table 2).

In the third trimester, we identified four patterns. The first (meats, eggs, fried snacks, and processed products) represented 14.08% of the consumption data; the second (cereals, legumes, vegetables/salads, fruit, and milk/dairy), 13.28%; the third (coffee and butter/margarine), 11.03%; and the fourth (sugars and sweets), 10.62% of the consumption data (Table 2). 

Pearson correlation between the dietary patterns of the first and third trimester were: Pattern 1: 0.035 (*p* = 0.6); Pattern 2: 0.023 (*p* = 0.7); Pattern 3: 0.085 (*p* = 0.2); Pattern 4: 0.051 (*p* = 0.4).

### 3.3. Dietary Patterns during Pregnancy and Birth Weight

The SEM presented satisfactory fit (*p* > 0.05). The first (meats, eggs, fried snacks, and processed food products) and fourth patterns (sugars and sweets) in the third trimester of gestation had a direct and significant negative impact, where the birth weight was reduced by 98.42 g (*p* = 0.009) and 92.03 g (*p* = 0.03), respectively (Table 3; Figure 3). There was no statistically significant difference between the pattern of dietary consumption in the first trimester and birth weight. Weight gain in pregnancy (β = 160; *p* = 0.04) and morbid complications during pregnancy (β = −145; *p* = 0.008) (anaemia, hypertension, diabetes mellitus) had a direct and significant impact on birth weight. Gestational weight gain had a direct and positive impact on birth weight (β = 160; *p* = 0.04). It was observed that for every 1 kg gained during pregnancy, there was a reported increase of 160 g in newborn birth weight. With regard to morbid complications in pregnancy, these directly and negatively impacted the weight of the child, reducing total weight at birth by 145 g (Table 3; Figure 3).

Loss over the 12 months of follow-up was 7.57% (*n* = 14). The most frequent reasons were those related to not finding the address provided by the pregnant woman to the family clinic (6; 25%), absence in one of the rounds of the study (8, 33.4%), and refusal to participate in the study (10; 41.6%) (Figure 1).

## 4. Discussion

This is one of the few studies conducted in Brazil that aims to identify patterns of dietary consumption during pregnancy and their relation to birth weight. It uses robust methodological modelling of follow-up and a convenient framework of multivariate data analysis, permitting the estimation of a series of relationships between variables. This study provides a convenient framework of multivariate data analysis and allows the estimation of a series of relationships between variables. In this sense, it was found that consumption of meat, eggs, fried/savoury foods, and processed food products (pattern 1) and consumption of sugar and sweets (pattern 4) in the third trimester had a direct and negative impact on birth weight, reducing it by 98.42 g and 92.03 g, respectively. 

Although some of these foods, such as meat and eggs, included in pattern 1, contain protein of high biological value that has a recognized physiological function in pregnancy, these foods are mainly fried because of the cultural influence in the population of the local municipality. Savoury, fried, and processed food products, contain high levels of saturated fat, sodium and cholesterol in their composition, which negatively affect the health of the individual and can also adversely affect newborn birth weight [29]. Furthermore, dietary pattern 3 (coffee and butter/margarine) during the third trimester significantly impacts birth weight in an indirect fashion, causing complications during pregnancy. Although this effect was small (β = −0.11) at the sample level, the overall risk could be significant. 

The third dietary pattern had an indirect association with anaemia. Observational studies show that coffee consumption is associated with anaemia [29]. However, the inhibitory effect of coffee on iron absorption can be partially reduced by the concurrent intake of vitamin C-rich foods and foods of animal origin [29]. Therefore, pregnant women should limit coffee consumption and avoid drinking coffee with meals [29].

Thus, the third dietary pattern may favour weight gain during pregnancy, increasing the likelihood of complications during pregnancy (hypertension, diabetes, anaemia) as well as reducing the availability of essential nutrients for the proper development of the foetus. 

The patterns of dietary consumption identified among the women in this study are similar to those found in the Brazilian population as a whole, which is explained by changes taking place in the population’s diet, and reflected in its epidemiological and nutritional profile at the different stages of life.

In Brazil, as in other countries, these changes are related to changes in lifestyle, which are expressed in the reduction in daily caloric expenditure; increased consumption of saturated fats, simple sugars, and processed foods; and reduced consumption of fruits and vegetables. This dietary pattern is a matter of concern at all stages of life, and during pregnancy it can be associated with poor maternal health conditions and compromise foetal growth and development. The lack or excess of nutrients during pregnancy can lead to morbid complications for both the mother and foetus, impacting the child’s health at later stages of life [1,8,29,30].

The four different dietary patterns identified in the first and third trimesters of pregnancy represented 50.45% and 49.03%, respectively, of the total consumption variance, indicating that the patterns extracted represent the eating habits of the population being studied.

The dietary intake profile of the pregnant women in this study predominantly consisted of foods rich in saturated fats, trans fats, cholesterol, sugars, and sodium, and poor in vegetables and fruit, which are good sources of fibre and micronutrients. This dietary profile, in a critical period of human development such as pregnancy, has consequences for the mother and foetus [31,32].

The causal paths of this relationship are still being studied, but the scientific evidence that reveals the influence of the maternal diet on the maternal–foetal gut microbiome stands out as promising [17,33]. It has been observed that diets with a high fat content are associated with reduced microbial diversity, while the consumption of fibres may be positively associated with microbial diversity [33], revealing that the diet during the intrauterine period can influence both the child’s immediate health and also health in subsequent stages of life [33], and may impact birth weight.

In the trimesters evaluated, we reported the consumption of fruits but at an extremely low frequency on a daily basis, accounting for only 9.11% in the first and 13.28% in the third trimester. This low consumption particularly restricts the availability of vitamins, minerals, and nutrients that directly impact appropriate weight and size at birth. Observational studies found that fruit consumption during pregnancy is conducive to adequate weight and size at birth and associated with a low incidence of preterm births [32,34].

One similar study involving pregnant women reported two patterns of food consumption: healthy (local preparations containing maize flour and maize meal, yams, fruits, vegetables, meat, and eggs) and unhealthy (sweetened drinks, ice cream, chocolate drinks, and soda) [15]. The healthy pattern was found to offer protection against low birth weight. In another study, maternal food patterns consisting of fruits, nuts, Cantonese desserts, and a varied diet (pasta, bread, vegetables, poultry, meats, fish, seafood, yogurt, and beans) was associated with high birth weight [35].

This study highlights the epidemiological importance of assessing the dietary intake of pregnant women during pregnancy; we identified the different consumption patterns during each trimester and associated these patterns to the different nutritional needs in these periods.

The need to avoid imbalanced food consumption must be emphasized, both in relation to excess and deficit. The adoption of healthy living habits, with a varied dietary intake, qualitatively balanced in nutrients and energy to meet each trimester’s nutritional needs, allows satisfactory physiological adaptations of the maternal organism and adequate foetal growth and development. Women who intend to become pregnant should maintain a healthy lifestyle, which includes maintaining a healthy weight, practicing physical activity, stopping tobacco and alcohol use, and adopting a balanced diet [36]. In addition to all these measures, pregnant women must maintain a balanced weight and adequate vitamin and mineral supplementation. All these factors contribute to maternal health and reduce the risk of negative outcomes for newborns [15]. In this sense, maintaining healthy eating habits is a requirement both before and during gestation. In Brazil, the Ministry of Health recommends following the guidelines of the Food Guide for the Brazilian Population as a practice of healthy eating [37].

Despite the need to follow the nutritional recommendations in this stage of life, evidence from a meta-analysis on micronutrient intake during pregnancy shows that women in this period ingest an insufficient quantity of micronutrients as opposed to the recommended values [38].

It is known that when pregnant women begin their pregnancy with an age-appropriate weight, their nutritional reserves (energy and nutrients) supply the foetus with its nutritional needs in the first trimester [39], which may be associated with the absence of a correlation between maternal dietary consumption in the first trimester and foetal growth and development in this period, as seen in this study.

However, some methodological issues of this study should be considered, especially regarding techniques of collection and analysis of dietary intake; sufficient measures were adopted to minimize the measurement biases that could be caused by recall, such as the use of a photo album containing pictures of food portions during the investigation of dietary consumption.

In this study, factor analysis with the principal component technique was adopted to identify dietary consumption patterns. Still, this method is not without criticism, mainly due to the subjectivity in determining the number of factors to be extracted. This restriction has been minimized with the adoption of methodological assumptions in the statistical model [40].

It is understood that losses in monitoring may contribute to selection bias in the investigated sample. However, on statistical analysis of the losses, it was noted that these did not differ in the sample throughout the monitoring period for some exposure variables, for example, pre-gestational anthropometric status, maternal age, income, education, smoking and alcohol consumption (data not shown in table), which indicates that the losses occurred at random. Of note, the losses were negligible (12%), so it was unlikely that they resulted in biased results.

Thus, longitudinal studies with a broad methodology for assessing consumption, suitable to the complexity of the Brazilian population’s diet, generate consistent information that can support the definition and implementation of health measures for ensuring adequate nutrition during pregnancy. The objective is to reduce maternal morbidity and mortality rates and guarantee ideal health and nutritional conditions for foetal growth and development.

It warrants mentioning that more studies are needed to clarify the causal paths involved in the relationship studied, especially investigations of the influence of the maternal dietary pattern on the infant gut microbiota and the impacts on perinatal outcomes. 

The results of this study indicate that unhealthy dietary consumption in the third trimester has negative effects on birth weight in two ways. Patterns one (meat, eggs, fried snacks, and processed foods) and four (sugars and sweets) directly impact birth weight by reducing it. The third pattern (coffee and butter/margarine), which also indicates inadequate intake, has the same impact on birth weight, however it acts indirectly, mediating the occurrence of increased morbidity.

These findings are relevant to the field of nutritional health for mothers and their infants, since food consumption is considered a modifiable factor; dietary and nutritional interventions during gestation, which are important determinants of birth weight, could contribute positively to the health of the mother and foetus.

## Figures and Tables

**Figure 1 nutrients-13-04054-f001:**
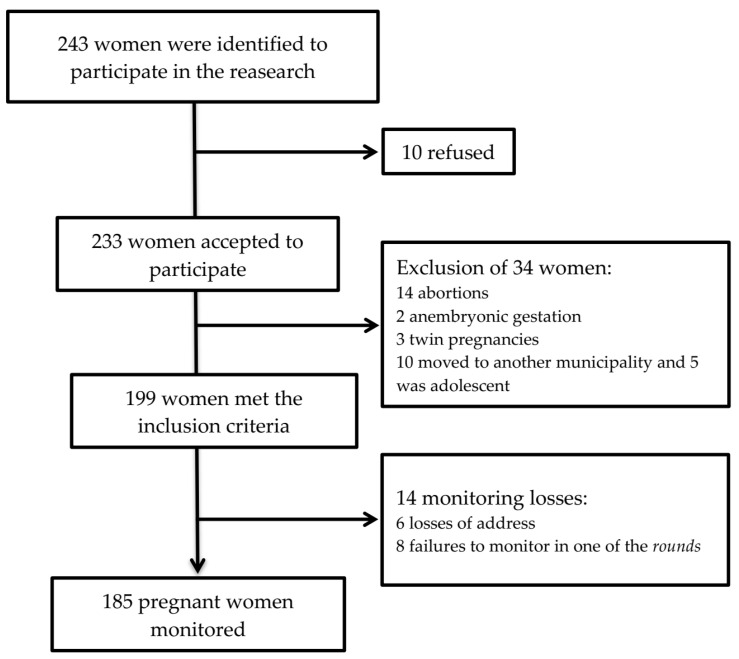
Monitoring flowchart of pregnant women. Cohort NISAMI. Santo Antônio de Jesus, Bahia, April 2012 and June 2013.

**Figure 2 nutrients-13-04054-f002:**
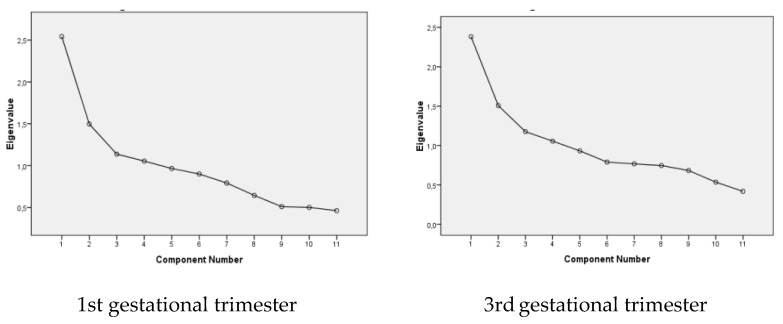
Cattel Graphic Test (scree plot) for dietary patterns identified in the first and third trimesters of pregnancy. Santo Antônio de Jesus, Bahia, 2012–2013.

**Figure 3 nutrients-13-04054-f003:**
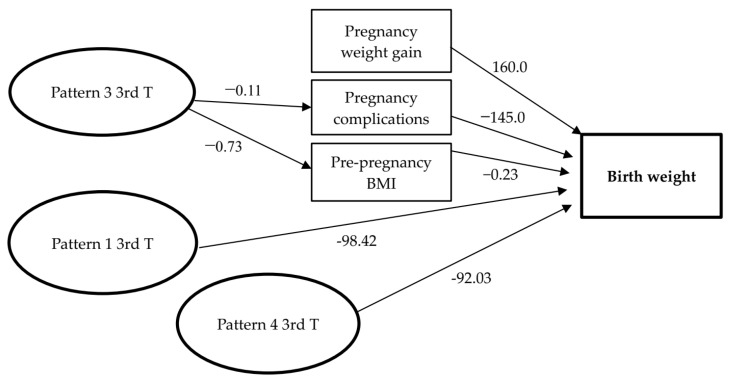
Structural equation modelling of the relationship between dietary patterns in pregnancy on birth weight. Cohort NISAMI. Santo Antônio de Jesus, Bahia, April 2012 and June 2013. Pattern 1 3rd T = Dietary Pattern 1 in the third trimester; Pattern 3 3rd T = Dietary Pattern 3 in the third trimester; Pattern 4 3rd T = Dietary Pattern 4 in the third trimester; Pregnancy Complications: (anaemia, urinary tract infection, gestational diabetes, hypertensive disorders); PP BMI: pre-pregnancy body mass index.

**Table 1 nutrients-13-04054-t001:** Mean of birth weight according to sociodemographic, biological, and anthropometric characteristics of mothers and their children, Santo Antônio de Jesus, Bahia, Brazil, 2012–2013.

Variables	Mean (SD)	*p*-Value ^1^	N	%
**Age (years)**				
<30	3320. 87 (483.1)		133	71.89
≥30	3393.13 (615.02)	0.39	52	28.11
**Maternal education level**				
≥Secondary education	3328.15 (595.13)		26	14.1
<Secondary education	3343.31 (512.11)	0.89	159	85.9
**Family Income**				
>1 Minimum salary	3371.67 (508.31)		139	75.1
≤1 Minimum salary	3249.07 (560.17)	0.16	46	24.9
**Antenatal visits**				
≥7 prenatal appointments	3376.87 (44.36)		109	58.9
<7 prenatal appointment	3307.38 (590.84)	0.56	76	41.1
**Smoker**				
Yes/ex-smoker	3375.00 (763.97)		16	8.6
No	3337.98 (497.25)	0.78	169	91.4
**Alcohol consumer**				
No	3369.55 (495.68)		161	87.0
Yes	3250.88 (658.89)	0.05	24	13.0
**Physical activity**				
No	3332.63 (520.91)		169	91.4
Yes	3431.56 (551.76)	0.47	16	8.6
**Pregnancy complications** (anaemia, urinary tract infection, gestational diabetes, hypertensive disorders)				
No	3376.87 (528.30)		136	73.5
Yes	3244.12 (499.78)	0.13	49	26.5
**Pre-gestational Anthropometric status**				
Healthy	3307.58 (482.85)		104	56.0
Overweight	3384.33 (570.25)	0.32	81	44.0
**Weight gain during pregnancy**				
≥10 kg	3424.36 (490.30)		80	56.8
<10 kg	3277.81 (540.03)	0.05	105	43.2
**Total**	3341.18 (522.32)		185	

^1^*p* value refers to the Student’s *t* test for equal variances.

**Table 2 nutrients-13-04054-t002:** Distribution of factor loads for the four components (dietary patterns) identified in the first and third trimesters. Santo Antônio de Jesus, Bahia, Brazil 2012–2013.

**Foods/Food Groups**		**Dietary Intake Patterns First Trimester ***		
	Pattern 1	Pattern 2	Pattern 3	Pattern 4
Cereals, roots, and tubers			0.621	
Legumes		0.637		
Fruit				0.861
Vegetables		0.695		
Milk and dairy products			0.654	
Meat and eggs		0.494		
Sugars and sweets	0.507			
Coffee	0.727			
Processed and industrialized foods	0.461			
Fats	0.737			
Fried Snacks			0.795	
% Accumulated variance	19.42	30.65	41.34	50.45
Eigenvalues	2.54	1.49	1.13	1.05
Cronbach’s Alpha	0.51	0.53	0.54	0.56
**Foods/Food Groups**		**Dietary Intake Patterns Third Trimester ****		
	Pattern 1	Pattern 2	Pattern 3	Pattern 4
Cereals, roots and tubers		0.667		
Legumes		0.649		
Fruit		0.429		
Vegetables		0.504		
Milk and dairy products		0.498		
Meat and eggs	0.599			
Sugars and sweets				0.526
Coffee			0.732	
Processed and industrialized foods	0.739			
Fats			0.722	
Fried Snacks	0.43			
% Accumulated variance	14.08	27.37	38.4	49.03
Eigenvalues	2.38	1.50	1.17	1.05
Cronbach’s Alpha	0.54	0.57	0.57	0.56

* 1st trimester: Extraction method—principal component analysis with Varimax rotation. Kaiser Meyer-Olkin (KMO) = 0.662. ** 3rd trimester: Extraction method—principal component analysis with Varimax rotation. KMO = 0.620.

**Table 3 nutrients-13-04054-t003:** Modelling of the structural equations to observe the direct and indirect effects between birth weight, maternal variables and dietary consumption patterns, Santo Antônio de Jesus, Bahia, Brazil, 2012–2013.

	Direct Effect		Indirect Effect		Total Effect	
	Coeff. (g) (CI 95%)	*p*-Value	Coeff. (g) (CI 95%)	*p*-Value	Coeff. (g) (CI 95%)	*p*-Value
Birth weight <- Pattern 1 (3rd T*)	−98.42 (−24.26; −172.59)	0.009			−98.42 (−39.74; −172.59)	0.013
Birth weight <- Pattern 4 (3rd T*)	−92.03 (−39.88; −165.30)	0.003			−92,03 (−38.43; −165.30)	0.003
Birth weight <- Weight gain	160.0 (36.47; 243.56)	0.004			160,0 (34.4; 243.56)	0.04
Birth weight <- Pregnancy Complications <- Pattern 3 (3rd T*)	−145 (−21.39; −211.45)	0.008	−0.11 (−0.04; 0.17)	0.001	−145.11 (−23.3; −211.0)	0.04
Birth weight <- PP BMI <- Pattern 3 (3rd T*)			−0.73 (−0.09; −0.14)	0.004	−0.73 (−3.6; 27.71)	0.13

Pattern 1 (3rd T*) = Dietary Pattern 1 in the third trimester; Pattern 3 (3rd T*) = Dietary Pattern 3 in the third trimester; Pattern 4 (3rd T*) = Dietary Pattern 4 in the third trimester; Complications: Pregnancy Complications; PP BMI: pre-pregnancy body mass index.

## Data Availability

The study did not report any data.

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
