# Peer review of "Associations between Maternal Dietary Patterns and Infant Birth Weight in the NISAMI Cohort: A Structural Equation Modeling Analysis"

_nutrients, 2021, doi:10.3390/nu13114054_

Round 1
Reviewer 1 Report
A very interesting and current article from the point of view of maternal nutrition in pregnancy. The authors argue how different nutritional patterns affect the birth weight of neonates. In addition, other variables can modulate this effect. However, some clarifications should be made for a better understanding of the article:
- In the material and methods, which means dynamic cohort (line 69)?
- The calculation of the sample size is very clear but the authors should specify on what basis they assume the difference of 100 g (line 75).
- Figure 1, the last part of 185 monitored pregnancies is repeated.
- I would like to congratulate the authors for the approximation of scores in section 2.3. However in the theoretical model (the figure caption is poorly labeled), the mediating variables could be modulating variables, affecting both the exogenous variables and the outcome, also, weight gain may be included and modeling PP BMI and feeding patterns. I suggest that you review the model and update it as the model is even more complex.
- A loading factor higher than 0.4 could give collinearity of the factors (line 166) how did you solve this?
- What about dietary pattern 2, and how does it affect the first trimester patterns? although there is no significant correlation, I think these arguments should be debated and reinforced in the text. The lack of correlations between first and third trimester nutritional patterns could be due to the type of rotation used in the PCA.
Minor comments:
- Decimals in the tables should appear with a dots.
- Check the grammar and some English spelling errors.
- The description of table 2 should be in order of appearance, lines 234/237 first.
- In the figure on page 8, the beta of PP BMI does not appear.
Author Response
Ms. Alina Guo,
Editor-in-Chief, Nutrients
We appreciate your suggestions and the possibility of publication in this journal. All changes made are highlighted in yellow in the manuscript file, however, here are some comments about the proposed changes:
Response to Reviewer 1 Comments
Point 1. A very interesting and current article from the point of view of maternal nutrition in pregnancy. The authors argue how different nutritional patterns affect the birth weight of neonates. In addition, other variables can modulate this effect. However, some clarifications should be made for a better understanding of the article:
- In the material and methods, which means dynamic cohort (line 69)?
Response: We called the dynamic cohort because the participants had different dates of entry into the study, but we chose to keep the prospective cohort in the text instead of the dynamic cohort
Point 2. The calculation of the sample size is very clear but the authors should specify on what basis they assume the difference of 100 g (line 75).
Response: When consulting the statistics responsible for calculating the sample, she informed that she used an average weight at birth of 3196 g and a standard deviation of 456 g [14], an error of 2%, a power of 80%, and an acceptable loss of 15%. Thus, we apologize for the misunderstanding and present the correct wording in the text.
Point 3. Figure 1, the last part of 185 monitored pregnancies is repeated.
Response: Revised and changed in figure.
Point 4. I would like to congratulate the authors for the approximation of scores in section 2.3. However in the theoretical model (the figure caption is poorly labeled), the mediating variables could be modulating variables, affecting both the exogenous variables and the outcome, also, weight gain may be included and modeling PP BMI and feeding patterns. I suggest that you review the model and update it as the model is even more complex.
Response: We agree that the theoretical model is much more complex than the one presented, however our focus was given only to the variables that were associated with the outcome in our study population. We reflect that figure 02 is very similar to figure 04 and therefore we consider it unnecessary to maintain the figure and the explanatory text that is already incorporated in the study results. If the editor considers maintenance necessary, please flag.
Point 5. A loading factor higher than 0.4 could give collinearity of the factors (line 166) how did you solve this?
Response: Factor analysis aims to identify a set of interrelated variables and therefore, a bit of multicollinearity is desirable (HAIR, 2009). We used Hair's (2009) reference to support our analyses, we had theoretical and statistical care in the design, respecting all the assumptions of the factor analysis. According to this author, factor loadings in the range of 0.30 to 0.40 are considered to meet the minimum necessary for the interpretation of the structure. Factor loadings > 0.7 are indicative of a well-defined structure. As we work with food consumption and we know from the scientific literature how complex it is to find a very well-defined structure with factor loadings > 0.7, we chose to work with the minimum load, a value adopted in most studies on food consumption.
In addition, we organize foods by food groups with similar nutritional characteristics to make the structures as homogeneous as possible.
Hair, J.F.; Anderson, R.E.; Tatham, R.L.; Black, W.C. Multivariate Data Analysis; 2009; Vol. 1; ISBN 9780138132637
We quote here articles that adopt similar factor loadings:
- da Mota Santana, J., Milagres, M. P., Dos Santos, C. S., Brazil, J. M., Lima, E. R., & Pereira, M. (2021). Dietary intake of university students during COVID-19 social distancing in the Northeast of Brazil and associated factors. Appetite, 162, 105172.
- Paknahad, Z., Fallah, A., & Moravejolahkami, A. R. (2019). Maternal dietary patterns and their association with pregnancy outcomes. Clinical nutrition research, 8(1), 64-73.
- Coelho, N. D. L. P., Cunha, D. B., Esteves, A. P. P., Lacerda, E. M. D. A., & Theme, M. M. (2015). Dietary patterns in pregnancy and birth weight. Revista de saude publica, 49, 1-10.
- Knudsen, V. K., Orozova-Bekkevold, I. M., Mikkelsen, T. B., Wolff, S., & Olsen, S. F. (2008). Major dietary patterns in pregnancy and fetal growth. European journal of clinical nutrition, 62(4), 463-470.
- Englund-Ögge, L., Brantsæter, A. L., Juodakis, J., Haugen, M., Meltzer, H. M., Jacobsson, B., & Sengpiel, V. (2019). Associations between maternal dietary patterns and infant birth weight, small and large for gestational age in the Norwegian Mother and Child Cohort Study. European journal of clinical nutrition, 73(9), 1270-1282.
- OLINTO, MTA. Padrões alimentares: análise de componentes principais. In: KAC, G., SICHIERI, R., and GIGANTE, DP., orgs. Epidemiologia nutricional [online]. Rio de Janeiro: Editora FIOCRUZ/Atheneu, 2007, pp. 213-225. ISBN 978-85-7541-320-3. Available from SciELO Books
Point 6. What about dietary pattern 2, and how does it affect the first trimester patterns? although there is no significant correlation, I think these arguments should be debated and reinforced in the text. The lack of correlations between first and third trimester nutritional patterns could be due to the type of rotation used in the PCA.
Response: According to Olinto (2007), rotation is intended to improve the interpretability of factors; however, it does not improve the degree of fit with the data. We tried the orthogonal and the oblique rotations. On the oblique we didn't have interesting theoretical solutions, some cross-loads. We chose Varimax as it better approximates the reality of the population data and better clarifies the factorial matrix for interpretation. In addition, we were guided by the scientific literature, several articles on food consumption with pregnant women used the Varimax rotation.
Minor comments:
Point 7. Decimals in the tables should appear with a dots.
Response. Accepted. Changed in tables
Point 8. Check the grammar and some English spelling errors.
Response. Accepted.
Point 9. The description of table 2 should be in order of appearance, lines 234/237 first.
Response. Accepted.
Point 10. In the figure on page 8, the beta of PP BMI does not appear.
Response: Accepted. Included value in the figure

Reviewer 2 Report
Title
- It is not clear the place where the research study was conducted.
Abstract
- The abstract is poorly written and should be revised to give more details on the following: (1) the study design (prospective or retrospective cohort study); (2) the date of sampling took place and the age of pregnant women; (3) the significant P-values; and (4) future directions of research related to this work. Also in Line 29-30, authors should clarify how inadequate dietary intake associated with negative weight birth, directly and indirectly?
Keywords
- The lists should be revised. I suggest the following keywords: pregnancy, infant, dietary patterns, birth weight; factor analysis; cohort; Brazil.
Introduction
- More comprehensive and recent studies should be provided to support the aim of the study. I do believe that several studies are missing (e.g., Clin Nutr Res. 2019, 8(1):64-73; Am J Clin Nutr. 2021, nqab340; PLoS One. 2016, 11(9), e0162285).
- This section is short. In Line 43-57: I would suggest authors to focus on the following questions: How does a pregnant woman's diet affect the infant birth weight? Is there a link between maternal diet and infant gut health? How does maternal diet influence the infant gut microbiota during pregnancy? How does gut microbiota cause obesity and related diseases during pregnancy? I would suggest authors to look at these recent papers (Br J Nutr. 2020 Mar 4,1-29; Microorganisms. 2020, 8(8):1119).
- The importance of the study must be explained at the end of this section. The authors should clarify why this study is of interest? Why it is important to conduct this research in Brazil?
Materials and Methods
- Line 69-73: Many details on the study design are needed. How pregnant women were recruited? What was the drop-out rate? How authors deal with missing values? It is not clear to me if pregnant women are followed over time and data about them is collected as their characteristics.
- Line 81-85: The inclusion criteria should be clearly defined.
- Figure 1: The term “one of the rounds” is vague and should be clarified.
- Line 98: “rounds” please refer to the above comment.
- Line 98-100: please include if possible the reliability of FFQ.
- Line 101-111; Line 114-118: Data collection should be described in much more details. It is not clear how data was collected from women?
- Line 113: The heading could be changed (e.g., data on the study variables).
- Line 132-138: "In the follow-up of the study". Please include the time period here.
- Line 144: Theoretical model” is not appropriate heading as used here- please change.
- Line 161: “Prevalence was determined to describe the categorical variables”- meaning unclear.
- Line 182: Nutritional data analysis should be sufficiently described.
Results
- I think more results should be added in Table 2 such as Varimax rotation. Also, graphic Scree plot should be included. The SEM in the first semester is missing. Please clarify.
Discussion
- This section should be qualified for more recent studies on a topic. Please refer to my comment in introduction.
Conclusion
- Clear and critical future research directions to guide future research should be provided.
References
- Ref # 22 is very old- please delete.
Author Response
Ms. Alina Guo,
Editor-in-Chief, Nutrients
We appreciate your suggestions and the possibility of publication in this journal. All changes made are highlighted in yellow in the manuscript file, however, here are some comments about the proposed changes:
Response to Reviewer 2 Comments
Point 1. It is not clear the place where the research study was conducted.
Response: In section 2.1 Study design, information was added about the place where the study was carried out.
Abstract
Point 2. The abstract is poorly written and should be revised to give more details on the following: (1) the study design (prospective or retrospective cohort study); (2) the date of sampling took place and the age of pregnant women; (3) the significant P-values; and (4) future directions of research related to this work. Also in Line 29-30, authors should clarify how inadequate dietary intake associated with negative weight birth, directly and indirectly?
Response: We considered all pertinent information and added to the summary.
Keywords
- The lists should be revised. I suggest the following keywords: pregnancy, infant, dietary patterns, birth weight; factor analysis; cohort; Brazil.
The suggestion about changing the keywords was accepted.
Introduction
Point 3. More comprehensive and recent studies should be provided to support the aim of the study. I do believe that several studies are missing (e.g., Clin Nutr Res. 2019, 8(1):64-73; Am J Clin Nutr. 2021, nqab340; PLoS One. 2016, 11(9), e0162285).
- This section is short. In Line 43-57: I would suggest authors to focus on the following questions: How does a pregnant woman's diet affect the infant birth weight? Is there a link between maternal diet and infant gut health? How does maternal diet influence the infant gut microbiota during pregnancy? How does gut microbiota cause obesity and related diseases during pregnancy? I would suggest authors tolook at these recent papers (Br J Nutr. 2020 Mar 4,1-29; Microorganisms. 2020, 8(8):1119).
- The importance of the study must be explained at the end of this section. The authors should clarify why this study is of interest? Why it is important to conduct this research in Brazil?
Response: The writing of the introduction has been revised in light of the suggestions given by the reviewer. Older evidence was removed and replaced by current ones, all the indicated references were also added, as well as the interlocution between food consumption pattern in pregnancy and infant microbiota. We emphasize that we did not delve into these relationships, since our study does not present results that reveal the influence of dietary patterns on the infant's intestinal microbiota, and that is why this discussion was placed only in the theoretical field.Materials and Methods
Point 4. Line 69-73: Many details on the study design are needed. How pregnant women were recruited? What was the drop-out rate? How authors deal with missing values? It is not clear to me if pregnant women are followed over time and data about them is collected as their characteristics.
Response: A paragraph was prepared in section 2.3 to better contextualize the methodological aspects of the research, the form of recruitment and follow-up of the cohort.
There were no lost data, only for other variables that were not dealt with in this study. We had 7.56% loss in the cohort (14 participants) and 10 refusals. These data are shown in Figure 1.
Point 5. Line 81-85: The inclusion criteria should be clearly defined.
Response: The text involving the research inclusion criteria has been reformulated to make it clearer and more defined.
Point 6. Figure 1: The term “one of the rounds” is vague and should be clarified.
Response: We included in item 2.2 an explanation about rounds for better understanding.
Point 7. Line 98: “rounds” please refer to the above comment.
Response: In item 2.2, what was considered as rounds was explained, as mentioned above.
Point 8. Line 98-100: please include if possible the reliability of FFQ.
Response: We consider the analysis of reliability to be important, however, in the project that originated this study, the reliability of the food frequency questionnaire was not included in the methodology. We used the validated and calibrated questionnaire according to Brito (2015).
Reference:
Brito, S.M. Retenção ponderal materna no pós-parto: Um estudo de coorte em município do nordeste brasileiro.Tese (Doutorado – Doutorado em Saúde Pública) – Universidade Federal da Bahia, Instituto de Saúde Coletiva., 2015. Available in: <https://repositorio.ufba.br/ri/bitstream/ri/28854/3/Tese%20Sheila%20Monteiro%20Brito.%202015.pdf>. Access in: 27.Oct.2021.
Point 9. Line 101-111; Line 114-118: Data collection should be described in much more details. It is not clear how data was collected from women? Line 113: The heading could be changed (e.g., data on the study variables).
Response: We mentioned in the text that the entire methodological procedure was described in previous articles (5,19), but we accepted the suggestion and created a paragraph in section 2.3 “Data Collect and Dietary intake assessments” for a better description of the data collection and changed the title of the section for Data on study variables.
Point 10. Line 132-138: "In the follow-up of the study". Please include the time period here.
Response: Suggestion accepted and incorporated into the text
Point 11 Line 144: Theoretical model” is not appropriate heading as used here- please change.
Response: We agree that the theoretical model is much more complex than the one presented, however our focus was given only to the variables that were associated with the outcome in our study population. We reflect that figure 02 is very similar to figure 04 and therefore we consider it unnecessary to maintain the figure and this discussion block, which is already incorporated in the study results. If the editor considers maintenance necessary, please flag.
Point 12. Line 161: “Prevalence was determined to describe the categorical variables”- meaning unclear.
Response: Suggestion accepted, the text was modified to make it clearer.
Point 13. Line 182: Nutritional data analysis should be sufficiently described.
Response: Suggestion accepted. A reference to a previous study was offered with all the methodological details of the treatment of the food consumption variable.
Results
Point 14. I think more results should be added in Table 2 such as Varimax rotation. Also, graphic Scree plot should be included. The SEM in the first semester is missing. Please clarify.
Response: In table 02, eigenvalues ​​of each pattern of the first and third trimesters of pregnancy and Cronbach's Alpha values ​​were included. Added footer with KMO values ​​and varimax rotation information. In addition to adding the Scree Plot chart.
In the SEM analysis, the dietary patterns of the first and third trimesters and the mediating variables were considered; however, the dietary patterns of the first trimester did not imply weight in the outcome, in addition, the adjustment indices were not adequate. We were trying a model that represented the theory, but also that respected the assumptions of the analysis, with satisfactory adjustment indices.
Discussion
Point 15. This section should be qualified for more recent studies on a topic. Please refer to my comment in introduction.
Response: Accepted
- Conclusion
Point 16. Clear and critical future research directions to guide future research should be provided.
Answer: These suggestions have been incorporated into the text of the conclusion.
References
- Ref # 22 is very old- please delete.
- Answer: This reference is classic, used in health services and in surveys with pregnant women to classify their BMI according to gestational age. It is a reference recommended by all national and international health agencies. And it is part of our protocol for pregnant women at the Ministry of Health of Brazil. https://bvsms.saude.gov.br/bvs/publicacoes/cadernos_atencao_basica_32_prenatal.pdf

Round 2
Reviewer 2 Report
No further comments.